# Simplicity is Key: An Unsupervised Pretraining Approach for Sparse Radio Channels

## Abstract

Unsupervised representation learning for wireless channel state information (CSI) reduces reliance on labeled data, thereby lowering annotation costs, and often improves performance on downstream tasks. However, state-of-the-art approaches take little or no account of domain-specific knowledge, forcing the model to learn well-known concepts solely from data. We introduce **Spa**rse pretrained **R**adio **Tran**sformer (SpaRTran), a hybrid method based on the concept of compressed sensing for wireless channels. In contrast to existing work, SpaRTran builds around a wireless channel model that constrains the optimization procedure to physically meaningful solutions and induces a strong inductive bias. Compared to the state of the art, SpaRTran cuts positioning error by up to 28% and increases top-1 codebook selection accuracy for beamforming by 26 percentage points. Our results show that capturing the sparse nature of radio propagation as an unsupervised learning objective improves performance for network optimization and radio-localization tasks.

## 1. Introduction

Deep learning for wireless communication and sensing networks is an active research area and achieves outstanding performance in tasks such as network optimization and localization (Zhang et al., 2019). However, due to limited data availability, these approaches often tend to overfit the training dataset, leading to poor generalization capabilities (Simeone, 2018). The reason is the unconstrained optimization in the large parameter space, which results in suboptimal solutions (Vapnik, 2013). Unsupervised representation learning has gained significant attention in domains such as natural language processing (Devlin et al., 2019;

[1]Anonymous Institution, Anonymous City, Anonymous Region, Anonymous Country. Correspondence to: Anonymous Author <anon.email@domain.com>.

Preliminary work. Under review by the International Conference on Machine Learning (ICML). Do not distribute.

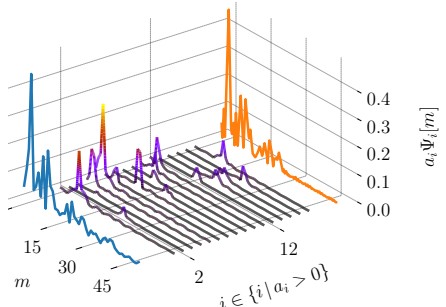

*Figure 1.* Example of learned sparse CSI decomposition, our unsupervised objective. The input signal (blue) is decomposed into a linear combination of basis functions (black) and subsequently recovered (orange).

Radford et al., 2018) and computer vision (Grill et al., 2020; Caron et al., 2021; He et al., 2020; Chen et al., 2020), often requiring fewer labeled samples for finetuning by making use of large amounts of unlabeled data in advance. Intuitively, unsupervised training acts as a regularizer that biases the network parameters towards regions of the parameter space whose local minima exhibit better generalization performance (Erhan et al., 2010). Consequently, recent works successfully adapted the paradigm to greatly reduce labeling effort for wireless tasks without degrading performance (Ott et al., 2024; Salihu et al., 2024; Alikhani et al., 2024; Guler et al., 2025).

While existing unsupervised methods for wireless applications have proven effective, they largely disregard well-known physical models of the wireless channel (Ott et al., 2024; Alikhani et al., 2024) or incorporate it implicitly via data augmentations (Salihu et al., 2024; Guler et al., 2025). Wireless systems are well suited for hybrid model- and data-driven approaches as there exists extensive domain-specific knowledge in the form of analytical approximations, yet the wireless channel includes complex stochastic processes which are infeasible to capture with mathematical models (O'Shea et al., 2019). Incorporating physical models in deep learning procedures improves generalization capabilities and reduces the dependency on large datasets (Shlezinger & Eldar, 2023). In this work, we show that explicitly incorporating a sparse wireless channel model into the unsupervised pretraining process improves down-

stream task performance. In other words, we introduce an additional bias that aids the pretraining process to find better solutions in parameter space. We draw inspiration from the compressed sensing (CS) framework for the architecture design and the pretraining objective. The central premise of CS is that sufficiently sparse representations reduce ambiguity, while in contrast, non-sparse representations typically contain numerous insignificant components, complicating both analysis and signal recovery (Donoho & Huo, 2001; Candes et al., 2006).

Our method, Sparse pretrained Radio Transformer (SpaR-Tran), is a physics-informed representation learning approach for CSI measurements. To match the sparse wireless channel model our architecture resembles partly a gated sparse autoencoder (SAE) (Rajamanoharan et al., 2024; Cunningham et al., 2023). However, unlike prior work considering SAEs, our goal is not to learn interpretable features but to aid the encoder towards a simpler solution by finding a sparse signal decomposition (see Fig. 1). In contrast to the state-of-the-art, SpaRTran uses single-links between antenna pairs instead of accumulating all available links of a system into a single datapoint. While sacrificing the ability to learn spatial characteristics between antennas in an unsupervised manner, this design choice brings an important advantage: The learned representations are system agnostic, i.e., they do not depend on the number of antennas or system topologies. This is an important step towards larger, more general wireless basis models that act on many different systems.

## 2. Related Work

SpaRTran is an unsupervised representation learning method for pretraining on wireless signals that integrates techniques from compressed sensing and dictionary learning.

**Unsupervised representation learning for CSI.** Supervised deep learning has advanced the state-of-the-art in terms of accuracy in tasks such as wireless positioning (Salihu et al., 2022; Liu et al., 2022; Zhang et al., 2023) and beam-management (Ma et al., 2023). Recently, unsupervised learning and self-supervised learning (SSL) has attracted significant attention in this context aiming to reduce the dependence on costly labels. The idea is to leverage cost-effective unlabeled channel measurements to pretrain a task-agnostic basis model also known as foundation model. Existing works transfer the pretraining objectives that have been established in domains such as computer vision or natural language processing to the wireless domain. Here, contrastive (Salihu et al., 2024) and predictive (Alikhani et al., 2024; Catak et al., 2025; Ott et al., 2024; Yang et al., 2025) methods have been studied as well as their combination (Pan et al., 2025; Guler et al., 2025). These approaches can be considered purely data-driven, learning the underlying struc-

ture of the wireless channel directly from measurements. To the best of our knowledge, hybrid model- and data-driven approaches have not been examined in this context so far.

**Compressed sensing** represents signals by a high-dimensional sparse vector in an overcomplete basis, assuming they arise from few latent factors. CS is widely used in wireless systems for source separation (Donoho, 2006; Candes et al., 2006), direction-of-arrival (DOA) estimation (Yang et al., 2018), and channel estimation (Berger et al., 2010). Classical recovery methods include convex relaxation (Chen et al., 2001; Tibshirani, 1996), greedy pursuit (Tropp & Gilbert, 2007), and sparse Bayesian learning (Malioutov et al., 2005; Stoica et al., 2011). Recent deep-learning approaches improve reconstruction fidelity while reducing complexity (Machidon & Pejović, 2023). Separately, SAEs (Cunningham et al., 2023; Bricken et al., 2023) enforce sparsity in high-dimensional latent spaces; Rajamanoharan et al. (2024) further decouple component selection from coefficient estimation to avoid shrinking of coefficients. We reinterpret the compressive optimization objective as a pretext task for unsupervised generative modeling, drawing on concepts from convex relaxation and SAEs to incorporate domain-specific knowledge into both, architecture and optimization.

**Dictionary learning** algorithms identify atomic features that sparsely represent underlying data, i.e., the dictionary is learned empirically from the signals themselves. This enables generalization across signal types and often leads to increased sparsity (Elad, 2010). A prominent example is the K-SVD algorithm (Aharon et al., 2006) which iteratively updates the dictionary atoms. SpaRTran can jointly learn the dictionary, increasing flexibility across waveforms and spatio-temporal patterns in cases where the theoretical dictionary may not approximate the signals sufficiently.

## 3. Problem Description

During a radio signal transmission, the electromagnetic wave interacts with the environment, i.e., the channel, which affects the signal, resulting in multiple propagation paths arriving at the receiver. The received signal $y(t)$ can be defined as $y(t) = h(t) * s(t) + w(t)$, where $s(t)$ is the transmitted signal, $h(t)$ the channel, $w(t)$ additive white Gaussian noise, and $*$ the convolution operator. The channel impulse response (CIR) $h(t)$ characterizes the radio transmission channel and can be modeled as

$$h(t) = \sum_{k=0}^{K-1} \alpha_k e^{-i\varphi_k} \delta(t - \tau_k), \qquad (1)$$

where $\tau_k$ is the signal transmission delay, $\alpha_k$ the magnitude and $\varphi_k$ the phase of the $k$-th propagation path of the transmitted signal. $\delta$ denotes the Dirac delta function and $i$ the imaginary unit. Eq. 1 describes the superposition of several

signals, originating from $K$ far field sources. In practice, $K$ is assumed to be unknown. The bandwidth-limited discrete channel measurement is modeled as

$$h[m] = \sum_{k=0}^{K-1} a_k \text{sinc}[m - \tau_k W] + w_n, \qquad (2)$$

where $W$ is the bandwidth of the system, $a_k$ are the complex-valued path coefficients, and $m \in \{1, \cdots, M\}$. From this, we derive the sparse channel representation. Assuming a set of $N$ potential signals $\boldsymbol{\psi}_l \in \mathbb{R}^M$ that form a basis, of which only $K \ll N$ effectively contribute to the received signal, we can rewrite Eq. 2 as

$$\boldsymbol{h} = \sum_{i=0}^{N-1} a_i \boldsymbol{\psi}_i + \boldsymbol{w}, \qquad (3)$$

where $|\alpha_i| > 0$ if the $i$-th signal is an active signal component, and $|\alpha_i| = 0$ otherwise. Note that we have replaced the sinc-function with a more generic notation $\boldsymbol{\psi}_i$. Defining the dictionary $\boldsymbol{\Psi} := [\boldsymbol{\psi}_0, \cdots, \boldsymbol{\psi}_{N-1}]$ allows (3) to be expressed more concise in matrix notation as

$$\boldsymbol{h} = \boldsymbol{\Psi}\boldsymbol{a} + \boldsymbol{w}, \qquad (4)$$

where $\boldsymbol{a} = [a_0, \cdots, a_{N-1}]^T$ is the sparse coefficient vector, and $\boldsymbol{\Psi}$ is a $M \times N$ dictionary matrix. Eq. 4 describes an underdetermined system of equations. As there is no unique solution, recovering the sparse channel requires solving the following optimization problem:

$$\min \|\boldsymbol{a}\|_0, \quad \text{s.t. } \|\boldsymbol{\Psi}\boldsymbol{a} - \boldsymbol{h}\|_2 \le \epsilon, \qquad (5)$$

where $\epsilon$ denotes the allowed reconstruction error due to noise. Eqs. (4) and (5) together describe the radio channel within the framework of compressed sensing (Donoho, 2006; Candes et al., 2006).

# 4. SpaRTran Training Pipeline

In general, we consider a set of unlabeled channel measurements $\mathcal{M}$. Our objective is to learn channel representations $\boldsymbol{z}$ that encode the environmental characteristics of the radio signal. To this end, we introduce a strong bias into the training process through both model architecture and loss function design. Our approach employs a data-driven encoder that generates a latent representation $\boldsymbol{z}$ and a decoder that reconstructs the input signal $\hat{h} \sim \mathcal{M}$ based on $\boldsymbol{z}$, using a hybrid data- and model-driven approach.

A transformer (TF) architecture (Vaswani et al., 2017) forms the backbone of the encoder. We employ a lightweight encoder-only TF with a depth of one and internal latent dimension of $N_{latent} = 512$, using 8 attention heads for the multihead attention mechanism. We interpret the complex values at the $m$-th timestep as a three-dimensional vector

consisting of the real and imaginary values as well as the magnitude $\hat{\boldsymbol{h}}_m = [\mathcal{R}(\hat{h}_m), \mathcal{I}(\hat{h}_m), |\hat{h}_m|]^T$. Similar to the state-of-the-art (Alikhani et al., 2024; Guler et al., 2025), we use non-overlapping patches of $\hat{\boldsymbol{h}}$ to construct our input embeddings $\boldsymbol{e}$. We combine windows of three time steps of the CIR into an input token $\boldsymbol{e}_m = [\hat{\boldsymbol{h}}_{3m-2}, \hat{\boldsymbol{h}}_{3m-1}, \hat{\boldsymbol{h}}_{3m}]^T$, ultimately resulting in 9 dimensions per token. We project each input token into the latent space of dimension $N_{latent}$ via a learned linear transformation to match the internal dimensionality of the TF-encoder. An ablation study comparing different TF configurations can be found in Appendix D.

## 4.1. Sparse Reconstruction Head

The sparse reconstruction head aims to find a sparse signal decomposition based on the representation $\boldsymbol{z}$. It uses a gating mechanism (inspired by Rajamanoharan et al. (2024)) and a phase generator. The former promotes the reconstruction to be sparse while the latter converts the real-valued output of the neural network to the complex-valued coefficients $\hat{\boldsymbol{a}}$. $\hat{\boldsymbol{a}}$ represents the reconstructed signal in terms of an overcomplete dictionary $\boldsymbol{\Psi}$, see Eq. 4. Fig. 2 shows the gating mechanism (yellow), the phase generator (green), and the dictionary (purple). Note that the learned representation $\boldsymbol{z}$ is itself not sparse.

We now discuss the gating mechanism in more detail. Approximating the $l_0$-norm with the $l_1$-norm tends to lead to non-optimal reconstruction. This is due to the fact that the sparsity penalty, i.e., the $l_1$-norm, can be reduced at the cost of reconstruction performance (Wright & Sharkey, 2024). Hence, our strategy for the estimation of $\hat{\boldsymbol{x}}$ follows the work of Rajamanoharan et al. (2024). The idea is to separately handle the selection of active atoms from the dictionary ($\boldsymbol{f}_{\text{gate}}$) and the estimation of the magnitude of the coefficients ($\boldsymbol{f}_{\text{coeff}}$). The encoder output is defined by

$$\hat{\boldsymbol{x}} = \boldsymbol{f}_{\text{coeff}}(\boldsymbol{z}) \odot \mathbf{1}(\underbrace{\boldsymbol{f}_{\text{gate}}(\boldsymbol{z})}_{\boldsymbol{\rho}_{\text{gate}}}), \qquad (6)$$

where $\mathbf{1}$ denotes the Heaviside step function, $\odot$ the Hadamard product and $\boldsymbol{\rho}_{\text{gate}}$ is the output of the gating stage before the binarization step. Fig. 2 shows the gating mechanism (yellow). Due to the binarization of the gating values, no gradient flows through this path of the network; see grey arrows in the yellow box of Fig. 2. Thus, an auxiliary loss promotes the detection of active atoms in $\boldsymbol{f}_{\text{gate}}$. The auxiliary loss measures reconstruction fidelity, but instead of $\hat{\boldsymbol{x}}$, it uses $\boldsymbol{\rho}_{\text{gate}}$ to reconstruct the signal. The dictionary should not be updated by the auxiliary reconstruction task. Hence, we prohibit the flow of the gradient accordingly (see grey dashed line in Fig. 2).

We now outline our extensions to the original method. Rajamanoharan et al. (2024) restrict the encoders output $\hat{\boldsymbol{x}}$ to real positive numbers. However, this assumption does not

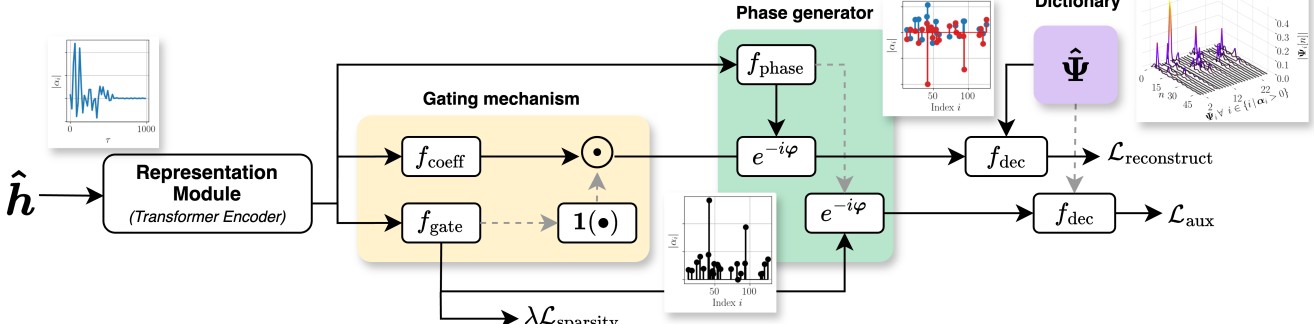

*Figure 2.* Overview of our unsupervised pretraining method - SpaRTran.

hold in our case, as our goal is to estimate complex-valued path coefficients $\hat{a}$. To address this, we interpret the outputs of $f_{\text{coeff}}$ and $f_{\text{gate}}$ as the magnitudes of the complex coefficients. This formulation allows us to suppress negative values via the gating mechanism without violating the underlying physical channel model. In addition, we introduce a third function $f_{\text{phase}}$ that generates the phases of the path coefficients. The final coefficients are then constructed as: $\hat{a} = \hat{x}e^{-if_{\text{phase}}(z)}$ and $\rho'_{\text{gate}} = \rho_{\text{gate}}e^{-if_{\text{phase}}(z)}$, where $i$ denotes the imaginary unit. The output of $f_{\text{phase}}$ is constrained to the interval $\pm\pi$ using a scaled tanh activation function. This leads to the following loss function:

$$\mathcal{L} := \underbrace{\|\tilde{h} - f_{\text{dec}}(\hat{a}, \hat{\Psi})\|_2^2}_{\text{reconstruction loss}} + \underbrace{\lambda\|\mathbf{1}(\rho_{\text{gate}})\|_1}_{\text{sparsity penalty}}$$
$$+ \underbrace{\|\tilde{h} - f_{\text{dec}}(\rho'_{\text{gate}}, \hat{\Psi}_{\text{frozen}})\|_2^2}_{\text{auxiliary loss}}, \quad (7)$$

with $f_{dec}(\hat{a}, \hat{\Psi}) = \hat{\Psi}\hat{a}$ (see Fig. 2, purple box). To enforce non-negativity, Rajamanoharan et al. (2024) employ ReLU activations for $f_{\text{gate}}$ and $f_{\text{coeff}}$. We observed that this can lead to a situation where certain dictionary atoms are never activated, i.e., their associated coefficients remain zero, resulting in no gradient updates, a phenomenon akin to the dying ReLU problem. To mitigate this, we use leaky ReLU activations (slope 0.01), ensuring that gradients can still propagate even for inactive units.

**4.2. Dictionary Composition**

There are two ways to obtain the dictionary. First, a theoretically derived fixed dictionary $\Psi \in \mathbb{R}^{M \times N}$ corresponding to the model we outlined in Sec. 3. Second, a learned dictionary $\hat{\Psi}$ that consists of jointly optimized parameters. We first detail the theoretical approach.

**Fixed Dictionary.** If the bandwidth $W$ of the considered signals is known it is reasonable to construct $\Psi$ from the theoretical model. In this case, the columns of $\Psi$ contain

shifted sinc-functions such that:

$$\Psi_{m,i} = \text{sinc}\left[m - i\frac{\tau_{max}W}{N}\right], \quad (8)$$

where $\tau_{max}$ is the last expected arrival of a significant signal component, which can be trivially determined by the number of time-steps $M$ of $\hat{h}$ and the sampling frequency $1/W$.

**Learned Dictionary.** If crucial system parameters, such as the signal bandwidth or the receiver's sampling frequency, are unknown, the construction of a dictionary that conforms to the theory is unfeasible. This may occur when using diverse crowd-sourced signals to train a large foundation model. In this case, SpaRTran can treat the dictionary as learnable parameters forming $\hat{\Psi}$. By normalizing the atomic entries to unit norm, they only determine the direction of the contribution, while $\hat{a}$ provides the amplitude and phase of the complex-valued signal component. However, jointly optimizing an unrestricted dictionary renders problem (5) NP-hard (Tropp, 2004) and the model-aided regularization effect is decreased (see also Sec. 5.4).

**4.3. Finetuning**

While SpaRTran learns representations for individual transmitter–receiver links, most tasks exploit correlations across multiple channels. We consider $N_r$ links recorded by an agent traversing the environment. We aggregate these into the channel state $H = [\hat{h}_1 \cdots \hat{h}_{N_r}]$, referred to as the CSI. Hence, for finetuning, we compute a representation for each available link and concatenate them to form a complete representation of the CSI. Our head is a 1D ResNet-style architecture starting with a $1 \times 1$ convolutional block that maps the output dimension $N_{latent}$ of the learned representation to 16 input channels. This is followed by a sequence of $N_{blocks,head}$ residual blocks. The $i$-th residual block outputs $16 * 2^{i-1}$ channels and uses a kernel size of $5 + 2(i-1)$, so both the channel width and receptive field grow with depth. A global average pooling layer, followed by a fully connected layer, maps the final feature representation to the estimate. We use $N_{blocks,head} = 4$ in our evaluations. An

ablation study comparing different head depths can be found in the appendix (Section D).

### 4.4. Theoretical Analysis

We treat the incoming signal as a time-dependent function $h(t)$ to be expressed in a reproducing kernel Hilbert space (RKHS) $\mathcal{H}$ with basis $\Psi = \{\psi_i\}_{i=0}^N$. For the sake of theoretical analysis, we treat the transformer encoder as an invertible operator $O : \mathcal{H} \to \mathcal{H}$ mapping from measured functions $h \in \mathcal{H}$ to latent representations $\tilde{h} = O[h]$. Our idea is to model the $\tilde{h}$ rather than $h$ itself to reach a more compact representation. We first characterize the error of a signal transformed by an invertible operator $O$. Note that all proofs for the theorems can be found in the appendix.

**Theorem 4.1.** *Let $\mathcal{H}$ be a reproducing kernel Hilbert space, equipped with a basis $\{\psi_i\}_{i=0}^N$. For any $h \in \mathcal{H}$ let the best n-term approximator be*

$$\sigma_n(h) = \min_{|I| \leq n} \|h - \sum_{i \in I} a_i \psi_i\|_{\mathcal{H}}.$$

*Also define the 1-atomic norm as*

$$\|h\|_{A_1(H)} = \inf\{\sum_i \|a_i\| : h = \sum_i a_i \psi_i\}$$

*Assume that there exists an exact recovery condition (ERC) (Tropp, 2004) such that the $O(n^{1/2})$ rate is optimal (Klusowski & Siegel, 2025)[1]). Then there exists a constant $C > 0$ such that for every $h \in \mathcal{H}$ we have*

$$\sigma_n(h) \leq C \frac{\|h\|_{A_1(\mathcal{H})}}{\sqrt{n}}.$$

*Let $O : \mathcal{H} \to \mathcal{H}$ be any invertible bounded linear operator with the standard operator norm $\|O\| = \sup_{\|h\|_{\mathcal{H}}=1} \|O[h]\|_{\mathcal{H}}$. Define the O-atomic norm of $f \in \mathcal{H}$ as*

$$\|h\|_{A_1^O} = \|O[h]\|_{A_1(\mathcal{H})}$$

*Then there exists an n-term representation $g_n$ in $\mathcal{H}$ such that*

$$\|O[h] - g_n\|_{\mathcal{H}} \leq C \frac{\|h\|_{A_1^O}}{\sqrt{n}}$$

*or, in the original space: $g_n = O[\tilde{h}_n]$*

$$\|h - \tilde{h}_n\|_{\mathcal{H}} \leq \|O^{-1}\| C \frac{\|h\|_{A_1^O}}{\sqrt{n}}$$

The result in Theorem 4.1 implies that if one tries to bound $K$ functions simultaneously,

$$\max_{0 \leq i \leq K} \|h_i - h_{i,n}\| \leq \|O^{-1}\| C \frac{\max_{0 \leq i \leq K} \|h_i\|_{A_1^O}}{\sqrt{n}}$$

---

[1]This is equivalent to asserting a generalised Jackson-type inequality over the codebook established by Temlyakov (2011).

preconditioning with $O$ improves upon the direct estimation if there exists a common substructure that allows us to reduce the coefficients in $\|h_i\|_{A_1^O}$ without overly affecting $\|O^{-1}\|$.

This is not unexpected: This is exactly what happens in signal smoothing where $O$ flattens singularities, or in differential preconditioning for PDEs. We provide an example of an analytically derived operator in Section B. Instead of using prior knowledge, we decide to learn $O$ with the option to fit it jointly with the dictionary $\Psi$ on a dataset of $\{h_0, \ldots, h_K\}$ such that the operator preconditions the entire set.

Specifically, we can construct such an operator as follows:

**Theorem 4.2.** *Let $\{h_i\}_{i=0}^K$ be a dataset of signals, and $\mathcal{H}$ be a Hilbert space with a (possibly infinite) basis $\{\psi\}_{j=0}^N$. Let $R = \max_{0 \leq i \leq K} \sum_{j=1}^N |a_{i,j}|$ and fix a nonempty index set $S \subset \{1, \ldots, N\}$. Define*

$$R_S := \max_{0 \leq i \leq K} \sum_{j \in S} |a_{i,j}| \qquad B := \max_{0 \leq i \leq K} \sum_{j \notin S} |a_{i,j}|$$

*If $B < R_S$ then there exists an operator $O$ such that*

$$\max_{0 \leq i \leq K} \|h_i - h_{i,n}| \leq C \frac{\max_{0 \leq i \leq K} \|h_i\|_{A_1^O}}{\sqrt{n}}$$

*which is strictly better than the original rate $C \frac{R}{\sqrt{n}}$*

*Remark* 4.3. The assumptions in Theorem 4.2 are quite technical, but can be boiled down to "One can sacrifice a more irrelevant subset $\bar{S}$ in favor of modeling a relevant subset $S$ with higher weight." In practice, this is a small assumption for an uninformed selection of the basis.

Of course, the "small index set" assumption for Theorem 4.2 is generally restrictive, but we can easily generalize the proof towards a low-rank assumption:

**Corollary 4.4.** *Assume there exists a rank $s$ subspace $S$, then the same bounds from Theorem 4.2 hold.*

*Remark* 4.5. The assumptions in Theorem 4.4 may seem strong at first, but in practice boil down to a weaker version of the assumptions necessary for compressed sensing: The real signal lives in a lower dimensional (function) space than the (noisy) measured signal

One important consequence of Theorem 4.4 is that we can find a smaller basis expansion, i.e., the weighted sum of basis functions in $\Psi$, by implicitly learning an infinite-dimensional dense rotation and diagonal scaling operator $O$. While simple finite operations at first appear to be sufficient for this, one has to note that the basis set we work with is generally infinite, meaning an orthogonal projection is not naively parameterizable. We decide to approximate $O$ with a TF encoder network by jointly minimizing the reconstruction error after applying $O$ together with the codebook $\Psi$.

*Table 1.* Fingerprinting (FP) performance for SparTRan and the baselines models for different amounts of labeled training data evaluated on the FH-IIS dataset (MAE / CE90 in meter).

| Method | 1% | 2% | 5% | 10% | 25% | 50% | 100% |
|---|---|---|---|---|---|---|---|
| **Masking** | 0.73 / 1.27 | 0.61 / 1.08 | 0.50 / 0.89 | 0.48 / 0.84 | 0.42 / 0.75 | 0.44 / 0.76 | 0.40 / 0.70 |
| **LWM** | 1.17 / 2.10 | 0.91 / 1.63 | 0.72 / 1.27 | 0.63 / 1.11 | 0.56 / 0.99 | 0.55 / 0.97 | 0.58 / 0.99 |
| **SWiT** | 2.33 / 4.24 | 2.09 / 3.81 | 1.90 / 3.46 | 1.79 / 3.29 | 1.76 / 3.22 | 1.65 / 3.01 | 1.51 / 2.75 |
| **Contra-WiMAE** | **0.52 / 0.93** | **0.48 / 0.86** | **0.45 / 0.80** | 0.43 / 0.76 | 0.39 / 0.70 | 0.42 / 0.74 | 0.39 / 0.69 |
| **SpaRTran** | 0.77 / 1.40 | 0.58 / 1.04 | 0.50 / 0.91 | **0.41/ 0.74** | **0.34 / 0.62** | **0.30 / 0.56** | **0.30 / 0.55** |
| **WiT (Sup.)** | 1.27 / 2.32 | 1.05 / 1.88 | 0.88 / 1.59 | 0.89 / 1.61 | 0.66 / 1.21 | 0.56 / 1.03 | 0.49 / 0.90 |

In short, our backbone transforms channel measurements to align with compressed sensing assumptions, biasing outputs toward simpler and more general solutions.

# 5. Evaluation

We evaluate localization via CSI fingerprinting (Secs. 5.1, 5.2) and codebook selection for beamforming (Sec. 5.3) — both leveraging full CSI complexity. In Sec. 5.4, we test our approach under several ablations and study the effect of varying values for sparsity penalty $\lambda$ and dictionary size $N$. For our experiments, we compare four SSL baselines (**SWiT** (Salihu et al., 2024), **Masking** (Ott et al., 2024), **LWM** (Alikhani et al., 2024) and **Contra-WiMAE** (Guler et al., 2025)) and a supervised method (**WiT** (Salihu et al., 2022)), each using a TF backbone. We evaluate all methods on three publicly available datasets: (i) **KUL**, a small controlled environment (Bast et al., 2020), (ii) **FH-IIS**, a larger and more complex environment (Stahlke et al., 2024), and (iii) **DeepMIMO** (Alkhateeb, 2019) for large-scale, diverse urban scenarios used in the codebook selection task. More detailed information on the baselines, training procedure, and the datasets can be found in Appendix C.

**Downstream Tasks.** We use two downstream tasks to evaluate the performance of SpaRTran:
(1) *CSI fingerprinting* maps positions to CSI measurements by exploiting their high spatial correlation. Due to multipath wave propagation, the CSI is typically highly characteristic per position, assuming a wide-sense static environment (i.e., negligible changes in the radio environment between training and inference) (Niitsoo et al., 2019; Stahlke et al., 2022). We use the mean absolute error (MAE) and 90th percentile of the cumulative error (CE90) to determine the positioning accuracy.
(2) *Beamforming* adapts phases/amplitudes across large phased array antennas to perform directed signal transmissions, mitigating high path losses and interference. Codebook selection for beamforming aims to select the optimal beam from a predefined codebook directly from channel measurements. This reduces the channel estimation overhead (Giordani et al., 2019). To assess the performance of

this classification task, we use the top-1 accuracy.

## 5.1. Localization in small data regime

To evaluate data efficiency, we train the methods on the FH-IIS dataset while using varying amounts of labeled data for finetuning, expressed as a percentage of the finetuning split (see Table 1). SpaRTran achieves the highest accuracy when at least 10% of the labeled samples are used, improving MAE up to 28% and CE90 up to 24% over the best baseline Contra-WiMAE, thereby demonstrating the effectiveness of our approach. For finetuning with $\leq 5\%$, however, SpaRTran performs worse than Contra-WiMAE. This is not surprising, as SpaRTran is pretrained on single links between antennas rather than the full CSI, preventing it to learn inter-channel correlations in an unsupervised manner. Furthermore, the compressive pretraining objective trades fine-granular, environment-specific signal patterns for a more general representation (see Sec. 4.4), reducing accuracy compared to less general methods that benefit from the fixed environment and system setup.

## 5.2. Localization under domain shift

Table 2 presents the results of wireless localization trained on pairs of scenarios, one used for pretraining and the other for finetuning. SpaRTran achieves the best accuracy in most cases. In particular, for off-diagonal results (i.e., domain shifts), SpaRTran achieves an average improvement of 15% in MAE and 33% in CE90, reaching MAE $\leq 0.071\,\mathrm{m}$ and CE90 $\leq 0.131\,\mathrm{m}$ even in the challenging URA-nLoS case. While all baseline methods exhibit a marked decline in performance under challenging non-line-of-sight (nLoS) conditions—i.e. when the most dominant, direct signal path is blocked—SpaRTran maintains a high accuracy. This highlights SpaRTran's superior ability to extract meaningful signal features beyond the dominant line-of-sight (LoS) path. Notably, in LoS cases, using the same pretraining and finetuning setup yields the best results for Contra-WiMAE. However, the significant degradation under domain shifts suggests that the method overfits the available data, whereas SpaRTran generalizes well due to its model-centric approach.

*Table 2.* FP performance across different system setups finetuned on 5 000 samples of the KUL dataset (MAE / CE90 in meter).

| Method | Pretrain-Set | DIS-LoS | ULA-LoS | URA-LoS | URA-nLoS |
|---|---|---|---|---|---|
| **Masking:** | DIS-LoS | 0.093 / 0.158 | 0.065 / 0.118 | 0.071 / 0.129 | 1.176 / 1.626 |
| | ULA-LoS | 0.081 / 0.139 | 0.067 / 0.116 | 0.071 / 0.127 | 1.094 / 1.615 |
| | URA-LoS | 0.087 / 0.156 | 0.068 / 0.118 | 0.073 / 0.131 | 1.176 / 1.671 |
| | URA-nLoS | 0.072 / 0.128 | 0.067 / 0.120 | 0.073 / 0.139 | 1.153 / 1.627 |
| **SWiT:** | DIS-LoS | 0.071 / 0.139 | 0.069 / 0.138 | 0.057 / 0.119 | 0.154 / 0.324 |
| | ULA-LoS | 0.076 / 0.146 | 0.068 / 0.136 | 0.058 / 0.119 | 0.140 / 0.303 |
| | URA-LoS | 0.070 / 0.138 | 0.068 / 0.136 | 0.057 / 0.119 | 0.156 / 0.327 |
| | URA-nLoS | 0.077 / 0.148 | 0.068 / 0.137 | 0.059 / 0.119 | 0.152 / 0.326 |
| **LWM:** | DIS-LoS | 0.083 / 0.144 | 0.084 / 0.151 | 0.075 / 0.137 | 0.244 / 0.454 |
| | ULA-LoS | 0.085 / 0.151 | 0.082 / 0.146 | 0.079 / 0.147 | 0.256 / 0.500 |
| | URA-LoS | 0.075 / 0.132 | 0.082 / 0.147 | 0.065 / 0.122 | 0.204 / 0.372 |
| | URA-nLoS | 0.094 / 0.164 | 0.092 / 0.163 | 0.083 / 0.151 | 0.224 / 0.418 |
| **Contra-WiMAE:** | DIS-LoS | **0.040 / 0.065** | 0.081 / 0.152 | 0.159 / 0.395 | 0.741 / 1.390 |
| | ULA-LoS | 0.098 / 0.164 | **0.043 / 0.069** | 0.070 / 0.131 | 0.403 / 1.039 |
| | URA-LoS | 0.077 / 0.140 | 0.069 / 0.122 | **0.040 / 0.077** | 0.128 / 0.245 |
| | URA-nLoS | 0.213 / 0.357 | 0.362 / 0.613 | 0.220 / 0.424 | 0.114 / 0.207 |
| **SpaRTran:** | DIS-LoS | 0.127 / 0.186 | **0.055 / 0.098** | **0.043 / 0.087** | **0.071 / 0.131** |
| | ULA-LoS | **0.065 / 0.114** | 0.058 / 0.100 | **0.054 / 0.110** | **0.075 / 0.139** |
| | URA-LoS | **0.072 / 0.122** | **0.048 / 0.085** | 0.051 / 0.100 | **0.080 / 0.145** |
| | URA-nLoS | **0.064 / 0.109** | **0.048 / 0.082** | **0.038 / 0.073** | **0.077 / 0.142** |
| **WiT (Sup.):** | | 0.074 / 0.135 | 0.071 / 0.135 | 0.059 / 0.108 | 0.196 / 0.387 |

*Table 3.* Top-1 accuracy (%) for codebook selection in the beamforming task. Finetuning was performed across task complexities defined by codebook sizes (16, 32, 64, 128) and varying proportions of labeled training data (1%, 2%, 5%, 10%, 25%, 50%, 100%). Results for Contra-WiMAE are taken from (Guler et al., 2025), all others are obtained by reproducing the results.

| Method | Codebook Size | 1% | 2% | 5% | 10% | 25% | 50% | 100% |
|---|---|---|---|---|---|---|---|---|
| **Masking** | 16 | 50.7 | 59.6 | 68.5 | 72.4 | 76.8 | 78.4 | 80.6 |
| | 32 | 31.6 | 40.6 | 53.5 | 67.6 | 74.9 | 78.1 | 81.2 |
| | 64 | 15.9 | 23.4 | 33.2 | 43.6 | 59.1 | 64.8 | 65.0 |
| | 128 | 9.4 | 10.1 | 13.0 | 20.8 | 33.0 | 40.4 | 42.1 |
| **SWiT** | 16 | **56.7** | 61.3 | 73.5 | 78.6 | 81.3 | 83.7 | 84.3 |
| | 32 | 40.0 | 50.6 | 71.1 | 76.4 | 77.2 | 80.0 | 81.5 |
| | 64 | 30.1 | 36.7 | 51.6 | 60.1 | 69.5 | 70.2 | 68.0 |
| | 128 | 18.6 | 22.6 | 25.3 | 36.5 | 44.6 | 47.6 | 41.2 |
| **LWM** | 16 | 40.3 | 51.0 | 68.7 | 72.5 | 77.4 | 79.1 | 81.2 |
| | 32 | 21.0 | 28.4 | 51.9 | 67.3 | 76.3 | 82.0 | 82.4 |
| | 64 | 6.8 | 12.5 | 38.7 | 61.3 | 72.2 | 67.8 | 69.4 |
| | 128 | 4.0 | 4.3 | 10.8 | 17.5 | 51.8 | 44.0 | 57.7 |
| **Contra-WiMAE** | 16 | 54.2 | 65.3 | 73.8 | 76.2 | 81.3 | 83.0 | 85.0 |
| | 32 | 39.0 | 53.0 | 64.8 | 77.0 | 84.0 | 85.4 | 85.6 |
| | 64 | 24.9 | 32.5 | 44.7 | 55.0 | 67.7 | 75.8 | 74.7 |
| | 128 | 15.0 | 18.3 | 25.0 | 28.8 | 42.5 | 51.4 | 31.3 |
| **SpaRTran** | 16 | 56.0 | **71.6** | **82.8** | **84.6** | **89.4** | **90.0** | **92.8** |
| | 32 | 38.7 | **73.6** | **87.6** | **88.9** | **90.4** | **92.9** | **93.7** |
| | 64 | **32.3** | **54.2** | **74.4** | **84.2** | **86.6** | **89.5** | **89.0** |
| | 128 | **23.8** | **25.7** | **39.9** | **62.9** | **70.5** | **74.5** | **71.8** |
| **WiT (Sup.)** | 16 | 56.4 | 67.3 | 74.3 | 80.1 | 82.3 | 84.3 | 85.8 |
| | 32 | **44.3** | 69.1 | 70.2 | 79.4 | 82.1 | 82.9 | 84.4 |
| | 64 | 31.3 | 49.0 | 52.3 | 60.4 | 73.3 | 75.5 | 77,6 |
| | 128 | 15.9 | 24.2 | 21.7 | 34.4 | 48.4 | 48.8 | 49.8 |

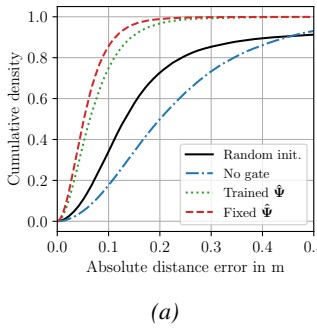
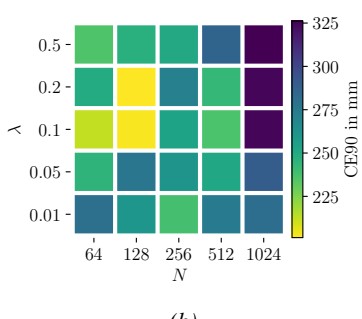
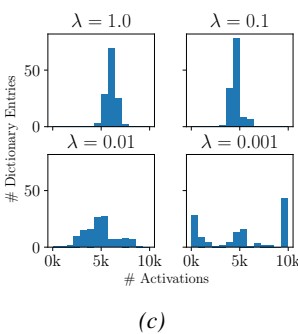

*(a)*          *(b)*          *(c)*

*Figure 3.* (a) shows the cumulative density of the wireless localization error under ablations, (b) shows the localization accuracy depending on the dictionary size $N$ and the sparsity coefficient $\lambda$ and (c) shows the distribution of number of activations on 10000 data points.

## 5.3. Beamforming

Table 3 shows the results for the beamforming downstream task, i.e., selecting the best beam ID in a predefined codebook to steer antenna gain in a specific direction to serve a mobile device. Again, SpaRTran consistently outperforms the compared methods. Notably, in one of the most challenging settings—a codebook size of 128 with fine-tuning on only 10% of the labeled training data—SpaRTran increases top-1 accuracy by 26 percentage points up to 62.9% relative to SWiT. Again, with a very low amount of labeled data (1%), SpaRTran's advantage diminishes due to the pretraining on single transmission links rather than full CSI. It is noteworthy that in the general picture none of the self-supervised baselines significantly outperforms the purely supervised method WiT. As the test environments were unseen during pretraining, this could indicate a low degree of generalization, occurring when the model overly adapts to the pretraining distribution. The model-induced bias in SpaRTran appears to make it less susceptible to this effect, improving its generalization capabilities.

## 5.4. Ablations

Figure 3a shows localization accuracy for the following ablations: no pretraining (randomly initialized backbone; black), no sparsity-inducing gating during pretraining (blue), and two regular pretraining variants with a learned dictionary (green) and a fixed dictionary based on the theoretical channel model (red). Both regular pretraining cases achieve a very high accuracy of CE90 $\leq 0.145$. The learned dictionary incurs a minor CE90 degradation of $0.003\,\mathrm{m}$ versus the fixed dictionary, so it remains competitive and is suitable when system configurations are unknown. SpaRTran reduces CE90 by $66.5\,\%$ relative to a randomly initialized backbone, demonstrating its effectiveness. Removing the sparsity-inducing gating worsens performance relative to the random initialization, underscoring the critical role of the sparsity assumption.

The parameter $\lambda$ balances a tradeoff between improved gen-

eralization with stronger sparsity and avoiding a systematic underestimation of the nonzero magnitudes (shrinkage) when $\lambda$ is too large (Wright & Sharkey, 2024). The dictionary size $N$ governs a second tradeoff: smaller dictionaries yield less coherent basis functions, simplifying identification of contributing basis functions, while larger dictionaries improve reconstruction fidelity (Donoho & Huo, 2001).

Figure 3c shows the distribution of number of activations per atom of a learned dictionary with $N = 128$ dependent on $\lambda$. In general, higher values of $\lambda$ (strong sparsity penalty) lead to a less spread out histogram, i.e., the atoms are activated with equal frequency, indicating an effective diversity of the learned dictionary. Note that when $\lambda$ is very small ($\lambda = 0.001$), some atoms dominate the representation, being activated constantly while others are stalled, an effect akin to mode collapse.

## 6. Conclusion

We presented SpaRTran, a hybrid model- and data-driven unsupervised method for learning task-agnostic radio channel representations. Our unsupervised objective aims to find a signal transformation that preconditions a sparse signal decomposition, i.e. reduces data ambiguity. This induces a strong inductive bias towards more general solutions. Furthermore, we use a gated SAE that integrates a channel model inspired by compressed sensing, reflecting the sparse model in the architectural design. Unlike existing methods, SpaRTran operates on individual radio links rather than full CSI matrices which decouples the model from specific system configurations, taking a step towards large generic radio foundation models. We presented results that show an improvement in generalization and overall performance over state-of-the-art approaches, achieving up to 28% reduction in positioning error and a 26 percentage point increase in the top-1 accuracy on codebook selection for beamforming.

## 7. Impact Statement

This paper presents work whose goal is to advance the field of machine learning. There are many potential societal consequences of our work, none of which we feel must be specifically highlighted here.

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

## A. Proofs

*Proof of Theorem 4.1.* Let
$$u := O[f] \in \mathcal{H}$$
then, by definition of $\|\cdot\|_{A_1(\mathcal{H})}$ we have for all $\varepsilon > 0$ a finite set of coefficients $c_i$ such that
$$u = \sum_{i \in I} c_i \psi, \quad \sum_{i \in I} |c_i| \le \|u\|_{A_1(\mathcal{H})} + \varepsilon = \|h\|_{A_1^O} + \varepsilon$$

Applying ERC to $u$ we get
$$\sigma_n(u) = \min_{J \subset [N], |J| \le n} \left\| u - \sum_{j \in J} a_j \psi_j \right\| \le C \frac{\|u\|_{A_1(\mathcal{H})}}{\sqrt{n}} \le C \frac{\|h\|_{A_1^O} + \varepsilon}{\sqrt{n}}$$

by definition, we have a
$$g_n = \sum_{j \in J_n} a_j \psi_j, \quad |J_n| = n$$
such that
$$\|u - g_n\|_{\mathcal{H}} \le C \frac{\|h\|_{A_1^O} + \varepsilon}{\sqrt{n}}$$

Finally, set $O^{-1}[g_n] = \tilde{h}_n$ to obtain
$$\|h - \tilde{h}_n\|_{\mathcal{H}} = \|O^{-1}[u - g_n]\|_{\mathcal{H}} \le \|O^{-1}\|\|[u - g_n]\|_{\mathcal{H}} \le \|O^{-1}\| C \frac{\|h\|_{A_1^O} + \varepsilon}{\sqrt{n}}$$

Now let $\varepsilon \to 0$ and we obtain our bound
$$\|h - \tilde{h}_n\|_{\mathcal{H}} \le \|O^{-1}\| C \frac{\|h\|_{A_1^O}}{\sqrt{n}}$$
$\square$

Assume the operator is diagonal in $\{\varphi\}$-Basis
$$O\left( \sum_j c_j \varphi_j \right) = \sum_{j=1}^N w_j c_j \varphi_j,$$
with $w_j > 0$.

Then the optimal diagonal weights are
$$w_j^\star = \begin{cases} 1/\alpha_{\max}, & j \in S, \\ M_{\text{large}}, & j \notin S, \end{cases}$$
where $M_{\text{large}}$ is chosen large enough not to affect $\|O^{-1}\|$.

From Theorem 4.1 we know that we need to compute $\|O^{-1}\|_{H' \to H}$ and $\|f\|_{A_1^O}$ for the operator bound.
$$\|O^{-1}\| = \max_j \frac{1}{w_j^\star} = \alpha_{\max}, \quad \max_{0 \le i \le K} \|f_i\|_{A_1^O} = \max_i \sum_j w_j^\star |a_{i,j}| = \frac{1}{\alpha_{\max}} \max_{0 \le i \le K} \sum_{j \in S} |a_{i,j}|.$$

Hence
$$M_{\text{diag}} = \alpha_{\max} \times \frac{1}{\alpha_{\max}} \max_{0 \le i \le K} \sum_{j \in S} |a_{i,j}| = \max_{0 \le i \le K} \sum_{j \in S} |a_{i,j}| \le \max_{0 \le i \le K} \sum_{j=1}^N |a_{i,j}| = \max_i \|f_i\|_{A_1}.$$

But since $|S| = s$, trivially

$$\max_{0 \leq i \leq K} \sum_{j \in S} |a_{i,j}| \ \leq \ \underbrace{s \, \alpha_{\max}}_{\text{generally strict}} \ << \ \max_i \sum_{j=1}^{N} |a_{i,j}|$$

whenever the original $\ell^1$-mass of $\{f_i\}$ is spread over many more than $s$ atoms. Concretely, if

$$\max_i \sum_j |a_{i,j}| = R$$

since, by assumption, $s << R/\alpha_{\max}$, then

$$M_{\text{diag}} \leq s\alpha_{\max} << R,$$

so we obtain a factor$\approx \frac{s \, \alpha_{\max}}{R}$ improvement in the non-asymptotic bound

$$\max_i \|f_i - f_{i,n}\| \leq C \, M_{\text{diag}}/\sqrt{n}$$

.

*Proof of Theorem 4.2.* Let $O$ be a diagonal operator with weights $w_i$ such that $O[h] = \sum w_i c_i \psi_i$. Let $w_i$ be defined as

$$w_i = \begin{cases} c & j \in S \\ 1 & j \notin S \end{cases}$$

with $c > 1$. Operators with this weighting have inverse norm $\|O^{-1}\| = \max(1/c, 1) = 1$ and

$$\max_{0 \leq i \leq K} \|h_i\|_{A_1^O} = \max_{0 \leq i \leq K} \sum_{j \in S} |a_{i,j}| + \frac{1}{c} \sum_{j \in S} |a_{i,j}| = B + \frac{1}{c} \max_{0 \leq i \leq K} \sum_{j \in S} |a_{i,j}|$$

Since, by assumption $B < R$ we may chose any $c$ large enough such that the residual

$$\frac{R_S}{c} < R - B$$

Specifically, choose

$$c > \frac{R_S}{R - B}$$

then

$$\max_{0 \leq i \leq K} \|h_i\|_{A_1^O} = B + \frac{R_S}{c} < B + (R - B) = R$$

In short, the resulting rate is (nonasymptotically) better than the original one. $\qquad\square$

*Proof of Corollary 4.4.* Let $M_{\text{diag}}$ be the advantage gained by a diagonal scaler. First apply an orthonormal transform $V$ projecting onto $s$ axes (via SVD or polar decomposition), then apply Theorem 4.2. This only adds the burden of back-projecting from $V$ space to the original space, i.e.

$$\max_i \|h_i - h_{i,n}\| \leq \|V^{-1}\| C \, M_{\text{diag}}/\sqrt{n}$$

which due to orthonormality yields

$$\max_i \|h_i - h_{i,n}\| \leq C \, M_{\text{diag}}/\sqrt{n}$$

$\qquad\square$

# B. Example Preconditioning Operator

Assume one wants to solve a Poisson differential equation using the Spectral method on the basis of Legendre Polynomials. Let's define the Poisson equation $-\Delta u(x) = f(x)$ for ground truth $f(x) = \frac{1}{\sqrt{1-x^2}}$ over $x \in (-1, 1)$ with $u(-1) = u(1) = 0$. Approximating $f$ with Legendre polynomials has slow decay for spectral methods due to the endpoint singularities. Preconditioning with Green's operator

$$O := \left( -\frac{d^2}{dx^2} \right)^{-1}$$

Then

$$O[f] = u$$

is smooth over $(-1, 1)$ and in fact has an analytic interior. Specifically Legendre coefficients in $f$ decay on the order of $n^{-1}$ due to the singularity, while coefficients in $u$ decay exponentially. For us this would mean

$$\|f\|_{A_1} := \inf \left\{ \sum_i |c_i| \Big| f = \sum_i c_i \psi_i \right\}$$

while

$$\|f\|_{A_1^O} = \|O[f]\|_{A_1} = \|u\|_{A_1}$$

by the convergence rate we get

$$\|f\|_{A_1^O} << \|f\|_{A_1}$$

The norm

$$\|O^{-1}\| = \| - \frac{d^2}{dx^2} \|$$

over $H^2 \cap H_0^1$ - The second order Sobolev space cut with the first-order subspace which vanishes on the boundary - is a fixed constant.

# C. Experimental Setup: Baselines & Datasets

## C.1. Baselines

Due to architectural differences and different data handling between the compared methods (for example, SpaRTran uses single links between antennas, whereas the baseline methods use full CSI) it is unfeasible to determine a common training length in terms of epochs or steps. Hence, to ensure a fair comparison, we fix the time for pretraining on a single Tesla V100 SXM2 GPU to 12 h and finetune all methods until convergence.

*Table 4.* Comparison of transformer hyperparameter.

| Method | $N_{latent}$ | $N_{hidden}$ | $N_{heads}$ | $N_{blocks}$ | #param |
|---|---|---|---|---|---|
| Masking | 512 | 1024 | 8 | 3 | 8.7 M |
| SWiT | 384 | 384 | 1 | 1 | 4.0 M |
| LWM | 64 | 256 | 1 | 12 | 1.3 M |
| Contra-WiMAE | 64 | 128 | 8 | 20 | 587 K |
| SpaRTran (ours) | 512 | 1024 | 8 | 1 | 2.6 M |

**SWiT**: Salihu et al. (2024) propose a joint embedding-based approach (Grill et al., 2020) called self-supervised wireless transformer (SWiT), that predicts the output of a momentum encoder, given different augmented views of the same input signal. The aim is to learn representations that are invariant to six randomly selected augmentations, diversifying the views.

**WiT**: Salihu et al. (2022) employs a compact TF model consisting of a single encoder block with single-head attention that is trained end-to-end in a supervised manner.

**Masking**: Ott et al. (2024) introduce a predictive objective for learning FP representations, in which masked portions of the input signal are reconstructed. During training, up to 50% of the input fingerprint is removed, forcing the model to learn spatiotemporal correlations between the multipath components (MPCs).

**LWM**: Alikhani et al. (2024) also use a masking strategy to pretrain the TF model. Here multiple patches of 15 time steps, of the input signal are gathered to be represented to the network as a token. They randomly select 15% of patches and, for those, replace 80% with a uniform MASK vector, 10% with random noise, and leave 10% unchanged.

**Contra-WiMAE**: Guler et al. (2025) combine a masked autoencoder with a contrastive pretraining objective. They generate positive samples by applying additive white gaussian noise to a CSI sample while treating all other channels in the batch as negative samples. The goal is to encourage the model to learn structural patterns via the masking objective and discriminative features via the contrastive objective.

### C.2. Datasets

*Table 5.* Comparison of dataset configurations.

| Dataset | $f_c$ [GHz] | $W$ [MHz] | $N_r$ | Area |
|---------|-------------|-----------|-------|------|
| KUL | 2.61 | 20 | 64 | $3\,\text{m} \times 3\,\text{m}$ |
| FH-IIS | 3.7 | 100 | 6 | $40\,\text{m} \times 30\,\text{m}$ |
| DeepMIMO | 3.5 | 20 | 32 | $>1\,\text{km}^2$ |

**KUL Dataset** (Bast et al., 2020): The dataset comprises four antenna configurations: distributed antennas (DIS-LoS), a uniform linear array (ULA-LoS), and a uniform rectangular array under both LoS (URA-LoS) and nLoS conditions (URA-nLoS). Each configuration contains 252,004 CSI samples with recording positions arranged in a grid-pattern with 5 mm distance. The channels are measured at 20 MHz bandwidth. We split the dataset randomly into 70 % for training, 10 % for validation, and 20 % for testing.

**FH-IIS Dataset** (Stahlke et al., 2024): This dataset contains CIR fingerprints collected using a 5G-FR1-compatible software-defined radio system (DL-PRS reference signal) with a bandwidth of 100 MHz. The recorded environment resembles an industrial hall featuring tall metal shelves, and a narrow corridor with large walls that introduce signal blockages and complex multipath propagation. The CSI is captured along a random walking trajectory of a person at a sampling rate of 6.6 Hz. Six base-stations are distributed along the perimeter of the localization area. The split sizes are as follows: 566,589 training, 141,639 validation and 593,022 test samples.

**DeepMIMO Dataset** (Alkhateeb, 2019): This is a synthetic dataset generated by a ray-tracing engine. We configure it to simulate a uniform linear array with 32 antennas at 20 MHz bandwidth, and 3.5 Ghz carrier frequency. We split the dataset into a pretraining and finetuning subset including 15 scenarios (*O1, Boston5G, ASU Campus, New York, Los Angeles, Chicago, Houston, Phoenix, Philadelphia, Miami, Dallas, San Francisco, Austin, Columbus, Seattle*) and 6 scenarios (*Denver, Fort Worth, Oklahoma, Indianapolis, Santa Clara, San Diego*) respectively. This yields the following split sizes: pretraining — 540,272 training and 135,075 validation samples; finetuning — 10,385 training, 1,481 validation, and 1,481 test samples.

We normalize the received signal strength of the FH-IIS and KUL datasets using a global normalization factor. This preserves the relative signal strength differences within the channel measurements, a crucial property for the downstream task of wireless localization. In contrast, DeepMIMO is normalized per CSI matrix to reduce sensitivity to outliers from large peaks that occur at short transmitter–receiver distances (signal strength scales roughly with $1/d^2$).

## D. Hyperparameter Analysis

Table 6 shows changes in localization accuracy in dependency of alterations in hyperparameters from the base model—i.e. architectural variations—and the corresponding number of parameters. In rows (a) we vary the depth of the finetuning head $N_{blocks,head}$ while the other hyperparameters remain fixed (see Sec. 4.3 for head network description). As the representation module handles only single links between antennas the spatial correlations between links in the full CSI must be learned solely by the head. This contrasts existing works which are less dependent on the expressivity of their head network as the spatial correlations are implicitly learned during pretraining. This dependency is clearly reflected in our results as the MAE gets consistently smaller with an increase in $N_{blocks,head}$ from $1.159\,\text{m}$ given $N_{blocks,head} = 2$ to $0.076\,\text{m}$ given $N_{blocks,head} = 5$. It should be noted that the number of parameters in the network increase drastically with larger $N_{blocks,head}$. This is why we choose $N_{blocks,head}$ to be the base configuration providing good results with a reasonable amount of parameters for the head network (1.4 million). Rows (b) provide insight into varying configurations of the representation module, i.e., the transformer encoder network. We alter the latent dimension $N_{latent}$ and the depth $N_{blocks}$

*Table 6.* Ablation Architectural Hyperparameter

| | $N_{latent}$ | $N_{blocks}$ | $N_{blocks,head}$ | MAE | CE90 | #param ($\times 10^6$) backbone | head |
|---|---|---|---|---|---|---|---|
| base | 512 | 1 | 4 | 0.093 | 0.177 | 2.61 | 1.40 |
| (a) | | | 3 | 0.128 | 0.240 | | 0.26 |
| | | | 2 | 0.159 | 0.303 | | 0.05 |
| | | | 5 | 0.076 | 0.149 | | 6.65 |
| (b) | 256 | | | 0.089 | 0.179 | 0.83 | |
| | 1024 | | | 0.211 | 0.409 | 9.31 | |
| | | 2 | | 0.064 | 0.126 | 4.71 | |
| | | 3 | | 0.113 | 0.223 | 6.81 | |

of the network. While a smaller latent of 256 has no relevant effect on the localization accuracy, doubling the size of it up to 1024 dimensions significantly increases the MAE by 227% from $0.093\,\text{m}$ to $0.211\,\text{m}$ and the CE90 by 231% from $0.177\,\text{m}$ to $0.409\,\text{m}$. As this comes with a radical increase in parameters up to $\approx 9$ million we think that the reduction in accuracy comes from overfitting the training data. The results for varying depths $N_{blocks}$ of the transformer encoder again indicate a tradeoff between overfitting and expressivity which is well balanced at $N_{blocks} = 2$. Nevertheless, we use $N_{blocks} = 1$ for our base configuration to reduce the amount of parameters making our model more lightweight than Masking and SWiT (see Table 4).

## E. LLM disclosure

We used large language models (LLMs) to assist with drafting and polishing the manuscript and for literature retrieval/discovery (e.g., to identify related work). Specifically, OpenAI GPT-4o and OpenAI GPT-5 were used to rephrase sentences for clarity and conciseness as well as search and summarize candidate related works. All LLM-generated text and suggested citations were reviewed, edited, and validated by the authors; final responsibility for the content, interpretations, and citation accuracy rests with the authors.

