# OpenReview forum: "Simplicity is Key: An Unsupervised Pretraining Approach for Sparse Radio Channels"
_ICML.cc/2026/Conference — Submitted to ICML 2026_

### Official Review · Reviewer_P1si · 2026-03-11

**Soundness:** 3
**Presentation:** 3
**Significance:** 3
**Originality:** 3
**Overall Recommendation:** 4
**Confidence:** 4

**Summary:**

This paper proposes **SpaRTran**, an unsupervised pretraining method for wireless CSI that builds sparse radio-propagation structure directly into both the architecture and the objective. A Transformer encoder produces single-link representations, which are decoded through separate **gating**, **coefficient**, and **phase** modules over an overcomplete dictionary. The main claim is that sparse, physics-informed pretraining yields more transferable representations than generic SSL objectives. Experiments on localization and beam-selection tasks show strong results overall, especially under domain shift and in moderate-to-large labeled-data regimes.

**Compliance With Llm Reviewing Policy:**

Affirmed.

**Key Questions For Authors:**

1. Can you compare against a **simpler sparse-coding / dictionary-learning baseline** without the Transformer backbone, and also against a Transformer with a **plain non-sparse reconstruction objective**, to better isolate the source of the gains?
2. Since SpaRTran is weaker in the **1–5% label regime**, how should practitioners interpret the claim of label efficiency?
3. Can you provide more concrete evidence linking the **theoretical preconditioning view** to the behavior of the learned encoder in practice?
4. Have you considered a controlled comparison between **single-link pretraining and full-CSI pretraining** within the same framework?

**Limitations:**

The paper should discuss limitations more explicitly, especially: weaker performance in the smallest-label regime, the limited baseline space for physics-informed alternatives, the loose theory–practice connection, and cases where **single-link pretraining may lose useful inter-link spatial information**.

**Strengths And Weaknesses:**

**Strengths**

1. The paper is **well motivated**: instead of directly importing generic SSL objectives into wireless learning, it uses a sparse channel prior that is well aligned with radio propagation structure.
2. The method is **coherent and reasonably novel**. The separation into atom selection, coefficient magnitude, and phase prediction is technically sensible for complex CSI, and the fixed-vs-learned dictionary design is practically useful.
3. The empirical results are **strong overall**. SpaRTran performs well on localization and beam-selection tasks, and the cross-domain results are particularly encouraging.
4. The ablations are useful and support the importance of the sparsity-inducing design.
5. The **single-link pretraining** perspective is an interesting design choice that may improve portability across different antenna configurations.

**Weaknesses**

1. The **baseline space is still limited**. The paper compares mainly to recent SSL baselines, but not to simpler physics-informed alternatives such as classical sparse coding / dictionary-learning pipelines or a Transformer pretrained with a simpler non-sparse reconstruction objective. This makes it hard to isolate what actually drives the gains.
2. The **small-label regime is a real weakness**. SpaRTran is not strongest at 1–5% labels, which weakens the claim of broad label efficiency, especially since this is arguably the regime where pretraining matters most in practice.
3. The **theory is only loosely connected** to the implemented model. The RKHS/preconditioning analysis is interesting, but it assumes a much cleaner structure than the nonlinear Transformer encoder actually used, so it reads more as motivation than explanation.
4. The **generalization story is not fully disentangled**: it remains unclear how much improvement comes from the sparse inductive bias, the single-link formulation, or other modeling choices.
5. Presentation is generally good, but the discussion of the **single-link vs. full-CSI tradeoff** could be more explicit.

---

> ### Author Rebuttal · Authors · 2026-03-30
>
> Thank you for reviewing our paper and recognizing the novelty in our approach as well as our strong experimental results.
>
> ## Weaknesses:
>
> > The baseline space is still limited.
>
> See answer to question 1
>
> > The small-label regime is a real weakness
>
> See answer to question 2
>
> > The theory is only loosely connected
>
> Theorems 4.1 and 4.2 act as a theoretical justification why a transformer that takes in an input signal $h(t)$ can produce a superior reconstruction compared to working on the native signal. Our core contribution remains the sparse-reconstruction objective for unsupervised pretraining.
>
> > The generalization story is not fully disentangled
>
> We would like to point out that we show an ablation that neglects the sparsity constrained in Section 5.4 (no Gates). Furthermore, as suggested by reviewer J897 we will add a simple autoencoder without sparsity constrained to the ablations to evaluate the impact of the single-link approach isolated. You can see the updated figure here: https://anonymous.4open.science/r/temp-0B8C/ablation_cdf.pdf .
>
> > single-link vs. full-CSI tradeoff
>
> Thank you for your suggestions of a more explicit discussion of the single-link vs. full-CSI trade-off we will consider this in the final version of the paper. Also see answer to question 4.
> Does the above clarifications resolve your concerns?
>
> ## Questions
> > 1. Can you compare against a simpler sparse-coding / dictionary-learning baseline without the Transformer backbone...
>
> We would like to note that sparse reconstruction serves a fundamentally different role in our framework compared to classical sparse coding methods. Rather than using it to directly extract signal parameters (e.g., angle of arrival) or for compression, we leverage it purely as a pretext task for representation learning.
> We use a transformer architecture backbone as this is in line with the current state-of-the-art. We did not consider another model as this could have a negative impact on the comparability of our method to the baselines. We would also like to point out that a non-sparse reconstruction objective is already explored in Section 5.4 of the paper (No gate). Thank you for raising this point—we will revise the relevant sections to improve clarity!
>
> > 2. Since SpaRTran is weaker in the 1–5% label regime, how should practitioners interpret the claim of label efficiency?
>
> Thank you for pointing out this lack of clarity. It is correct that our approach does not outperform the Contra-WiMAE baseline in the very low data regime (<10%) evaluated on the FH-IIS dataset. Nevertheless, SpaRTran achieves best performance in most of the cases in the very low data regime on the codebook selection task and shows superior domain shift capabilities while seeing only 5000 CSIs ($\approx 2\%$) in case of the KUL dataset.
>
> The weaker performance for the FH-IIS dataset can be attributed to the fact that SpaRTran processes only single links during pretraining and therefore must learn cross-link patterns exclusively during finetuning—which requires a certain amount of labeled data. This reflects a deliberate design trade-off: by restricting pretraining to single links, we significantly improve generalizability across diverse scenarios, at the cost of reduced performance in extremely low-label settings for tasks that depend on cross-link information. We will clarify this trade-off in the revised manuscript.
>
> > 3. Can you provide more concrete evidence linking the theoretical preconditioning view to the behavior of the learned encoder in practice?
>
> The proof is a pure existence proof: It is not even clear that this is the optimal operator or whether it is unique. The proof only states "There exists at least one such operator." If this were the optimal and unique operator, then in the limit of infinite parameters one would find exactly this operator (via the universal approximation theorem). Before that, one cannot say, just as I cannot say how close, e.g., a ResNet is to the optimal "infinite data, infinite model" limit.
>
> > 4. Have you considered a controlled comparison between single-link pretraining and full-CSI pretraining within the same framework?
>
> We appreciate this suggestion, and the proposed extension is indeed feasible in principle. However, we would like to emphasize that our key contribution—leveraging sparse signal decomposition as a pretext task—naturally operates on single-link measurements. The domain-specific knowledge induced through our approach captures the physical properties of individual wireless links but does not inherently model correlations between antennas. Since this novel pretraining paradigm represents the central contribution of our work, we intentionally focused our evaluation on single-link processing to isolate and validate its effectiveness. That said, extending SpaRTran to incorporate cross-antenna information is an interesting direction for future work, and we thank the reviewer for this valuable suggestion.

---

> > ### Author Rebuttal · Reviewer_P1si · 2026-04-03
> >
> > Responses in the rebuttal are helpful in clarifying certain aspects of the design and contribution. I would however keep my score as before, considering responses to questions 2 and 4.

---

### Official Review · Reviewer_MDYm · 2026-03-11

**Soundness:** 2
**Presentation:** 2
**Significance:** 3
**Originality:** 3
**Overall Recommendation:** 4
**Confidence:** 3

**Summary:**

This paper introduces SpaRTran (Sparse pretrained Radio Transformer), a hybrid model and data-driven unsupervised learning approach for wireless channel representation. The method combines compressed sensing principles with a gated sparse autoencoder architecture to learn task-agnostic radio channel representations. Unlike existing methods that operate on full CSI matrices, SpaRTran processes individual transmitter-receiver links, making it system-agnostic. The authors demonstrate improvements over state-of-the-art approaches on two downstream tasks: wireless positioning (fingerprinting) and beam selection for beamforming.

**Compliance With Llm Reviewing Policy:**

Affirmed.

**Final Justification:**

I would like to begin by thanking the authors for their thorough and detailed responses. Their clarifications have significantly improved my understanding of the evaluation setup and the motivations behind the methodological choices made. I also appreciate their commitment to incorporating numerous clarifications and additional details that were missing, at least from my perspective, in the original manuscript.

As a result, I am raising my score to weak accept.

The paper clearly demonstrates value: the proposed approach is well-grounded and opens promising directions toward scalable representation construction. The model architecture and training procedure reflect sound intuitions and thoughtful design choices.

The evaluation and comparison with state-of-the-art baselines are now clearer, and the authors have provided satisfactory justifications for their downstream head selection strategy.

On this last point, I understand the objective of demonstrating optimal task performance for each encoder, which is certainly a relevant goal. However, I remain concerned that this may not constitute a fully fair comparison in the context of foundation models. Established embedding benchmarks such as MTEB (https://huggingface.co/spaces/mteb/leaderboard), for instance, enforce a consistent evaluation protocol by using the same decoder across all compared models. While this approach has its own limitations, it provides a controlled basis for comparison across diverse tasks, independently of downstream architecture choices.

In the current setup, one could potentially identify alternative downstream head architectures that improve performance for competing baseline models, possibly surpassing the proposed approach. What conclusions could be drawn in such a scenario? This is why I would advocate for controlled comparisons with fixed downstream architectures, which would more directly and fairly assess the intrinsic quality of the learned representations. That said, I acknowledge that this is not always feasible or relevant, and that any fixed evaluation protocol carries the risk of inadvertently favoring specific encoder designs.

Regarding the notion of simplicity: I acknowledge the authors' response, though I still believe that alternative terminology might more accurately capture the intended meaning and better convey the practical benefits of the approach to the reader.

**Key Questions For Authors:**

- 1. Could you provide complete details on which datasets are used for pretraining each model in your experiments? Specifically, for the KUL and FH-IIS evaluations, what data is used for pretraining SpaRTran and all baseline methods? What is the quantity of pretraining data in each case?

- 2. Since SpaRTran processes individual transmitter-receiver links rather than full CSI matrices, please clarify: Is a single transformer encoder trained and then applied to all $N_r$ links, or are separate encoders trained for each link? If a single encoder is used, how do you account for the fact that it observes $N_r$ times more samples during pretraining compared to baselines that process full CSI matrices? The time for pretraining is fixed to 12h on a GPU, but it is not clear if all models reach this limit or not.

- 3. Could you provide a fair comparison of model complexity between SpaRTran and ContraWiMAE that includes the downstream decoder (1.4M parameters for SpaRTran)? Additionally, please discuss computational costs (training time, inference time, memory requirements) for both methods. This is particularly important given your paper's emphasis on "simplicity."

- 4. For the ContraWiMAE results on fingerprinting tasks (KUL, FH-IIS), which downstream model architecture is used? Only the linear probe mentioned in the DeepMIMO experiments, or also ResNet-based models like those used for SpaRTran? If only linear probing was used, could you provide results with ResNet downstream heads for fair comparison, or alternatively, evaluate SpaRTran with linear probing on these tasks?

**Limitations:**

Yes.

**Strengths And Weaknesses:**

**Strengths:** The paper presents several interesting contributions to wireless channel representation learning. The physics-informed approach explicitly incorporates domain knowledge through a sparse channel model based on compressed sensing theory, providing a principled foundation for the architecture design. The system-agnostic nature of processing individual links rather than full CSI matrices represents a step toward more generalizable wireless foundation models. The experimental validation is comprehensive, demonstrating strong performance across multiple datasets (KUL, FH-IIS, DeepMIMO) and tasks, with particularly good results under domain shift scenarios.


**Weaknesses:** Despite its contributions, the paper suffers from numerous clarity issues, missing explanations, and insufficient detail that hinder full comprehension and reproducibility. Notably:
- Section 4 describes the SpaRTran architecture but fails to adequately specify the implementation of key functions $f_\mathrm{coeff}$, $f_\mathrm{gate}$, and $f_\mathrm{phase}$. While Leaky ReLU activations are mentioned for $f_\mathrm{coeff}$ and $f_\mathrm{gate}$, it remains unclear whether these are simple MLP layers or more complex structures. This ambiguity extends to the auxiliary loss formulation, where the use of Leaky ReLU may introduce negative terms that are conceptually problematic.
- The theoretical analysis in Section 4.4, particularly Theorem 4.4, attempts to provide mathematical justification for the approach but lacks a clear connection to the actual implementation. There is no empirical evidence that the transformer encoder actually approximates the operator O described in the theorems, nor is the assumption of invertibility validated. In its current form, this theoretical section adds limited value to the paper and might be better suited for an appendix as speculative theoretical motivation rather than rigorous justification.
- The evaluation section presents the most significant concerns. First, except for DeepMIMO, the paper does not specify which data is used for pretraining the baseline models or the quantity of pretraining data employed. Second, since SpaRTran processes individual links rather than full CSI matrices, critical questions arise about training procedures: Is the same frozen encoder applied multiple times to generate different link representations, or is a separate encoder learned for each link? If a single encoder is used (as presumably intended), how are training samples counted? The encoder would observe $N_r$ times more samples than baseline methods during pretraining, making direct comparison problematic without proper accounting.
- As a note, for ContraWiMAE, results on DeepMIMO are taken from v1 of the reference paper rather than v2. From the used results, it seems that ContraWiMAE use linear probing. Is this also true for other tasks, or do evaluation protocols differ across methods? In this context, the parameter count comparison in Appendix C is misleading as it excludes the downstream decoder (1.4M additional parameters for SpaRTran, which is a bit counter intuitive as the paper title suggest a “simple” approach). More fundamentally, different representation learning methods are not compared on equal footing, as they use different downstream architectures and training procedures. SpaRTran's downstream head is particularly large, which could compensate for weaker representations and confound the comparison.
- While Table 2 suggests good generalization properties, the evaluation is restricted to specific database pre-training. The encoder does not demonstrate truly general-purpose representation learning; or at least, the current evaluation and explanations provide insufficient evidence to judge this.
- The ability to learn the dictionary (rather than using a fixed theoretical one) seems to contradict the goal of creating a foundation model with general representation capabilities. For a truly general foundation model, I would prefer to see a CSI embedding model trained once and functioning across a wide variety of contexts. Pretraining could leverage synthetic channels and statistical models, using the theoretical dictionary (Equation 8) with larger $W$ and$\tau_\mathrm{max}$ values to cover more cases. This would require standardizing input channels through preprocessing but would better align with the foundation model paradigm and the claim of increased generalization.
- Minor notation and presentation issues:
    - Equation (2): The noise term should be $w$, not $w_n$.
    - The notation $\psi_l$ could adopt the indexing from Equation (3) using index $i$, as the index $I$ is not meaningful in this context.
    - Using index $i$ for the summation in Equation (3) is unfortunate since $i$ already denotes the imaginary unit.
    - The paragraph discussing Equation (3) incorrectly refers to $\alpha_i$ instead of $a_i$.

Now regarding the specific items of the review:
- **Soundness:** The paper's technical soundness is mixed. While the compressed sensing foundation is well-established and experimental evaluation spans multiple datasets and tasks, critical issues undermine confidence. The theoretical analysis (Section 4.4) lacks empirical validation as there is no evidence that the transformer encoder approximates the operator $O$ or satisfies the assumed conditions. The experimental setup raises fairness concerns: pretraining data specifications are incomplete and link-level processing means SpaRTran observes $N_r$ times more samples than baselines. Baseline comparisons use inconsistent evaluation protocols with different downstream architectures.

- **Presentation:** The paper suffers from significant presentation weaknesses. Critical architectural details are missing (functions $f_\mathrm{coeff}$, $f_\mathrm{gate}$, and $f_\mathrm{phase}$ are not fully specified). The theoretical section (4.4) is poorly connected to implementation. Minor notation errors appear. Essential experimental details are absent: pretraining data specifications, whether the same frozen encoder is reused for all links, exact preprocessing steps, and more generally the training hyperparameters. The parameter count comparison (Appendix C) misleadingly excludes the 1.4M-parameter downstream decoder. The paper requires substantial revision for clarity and reproducibility.

- **Significance:** The paper addresses a relevant problem and demonstrates practical improvements, particularly in domain shift scenarios (Table 2). However, significance is limited by restricted evaluation scope and incomplete validation of key claims. The "system-agnostic" claim lacks validation through cross-system transfer experiments. While the physics-informed approach is interesting, the limited scope restricts potential impact.

- **Originality:** The paper's originality is moderate. The core contribution of combining compressed sensing with transformer-based representation learning for wireless channels and a gated sparse autoencoder for pre-training, represents a novel integration of established techniques. The gated sparse autoencoder is adapted from Rajamanoharan et al. (2024) with modifications for complex-valued channels. The link-level processing differs from prior work processing full CSI matrices, though advantages are not fully demonstrated. The sparse reconstruction pretext task offers a fresh perspective for the wireless domain. Overall, the work provides an interesting combination of existing techniques rather than fundamental conceptual advances.

---

> ### Author Rebuttal · Authors · 2026-03-30
>
> We sincerely thank you for your thoughtful and extensive evaluation of our work. We will revise the manuscript to improve reproduciblity and presentation issues.
>
> ## Weaknesses:
>
> > Implementation of f_coeff, f_gate and f_phase
>
> The functions $f_{coeff}$, $f_{gate}$, and $f_{phase}$ are implemented as lightweight MLPs. Regarding negative coefficients, these correspond to a 180-degree phase shift of the signal, which remains consistent with our channel model.
>
> > Cited Results from Guhler et. al.
>
> We have updated our table after reviewing the new results. Notably, Guler et al. report best performance in the low-data regime (≤10%) using linear probing—precisely the regime our evaluation targets. (Also see answer to Qu. 4). As our goal is to evaluate each pretraining objective's best achievable performance, different optimal fine-tuning strategies may naturally apply.
>
> ## Questions:
>
> **Question 1: Dataset details.**
> KUL Dataset: We used a 70% split of the full dataset for pretraining (176,402 CSI samples). For finetuning, we drew a random subset of 5,000 samples from this same pretraining split. FH-IIS Dataset: The pretraining set contains 566,589 CSI samples from the training split. We drew the finetuning subsets (1–100%) from this same pool.
>
> **Question 2: Single-link vs full CSI comparison.**
> We train a single encoder applied to all links. During finetuning, this encoder generates a representation per link, which are then concatenated before passing to the downstream head. We will clarify this in Section 4.3.
>
> You are correct that our model requires $N_r$ times more forward passes per epoch due to the link-wise splits. We carefully considered how to ensure a fair comparison. One option is to reduce CSI measurements by a factor of $N_r$ to equalize computational cost, but this would limit the channel information available to our model, placing it at a disadvantage relative to baselines observing the full context.
>
> **Question 3: Comparison SpaRTran vs. ContraWiMAE.**
> You correctly identified that SpaRTran's performance is influenced by downstream head capacity, as explored in Section D. We deliberately preserved the original baseline implementations as modifying baseline architectures could introduce confounding factors, obscuring whether performance differences stem from the pretraining objectives. We believe these architectures are already optimized for their respective methods. Nonetheless, we recognize this is a valuable point and performed the experiments that you asked for, see https://anonymous.4open.science/r/temp-0B8C/large_head_results.md.
>
> Overall, the larger head leads to worse performance than the original head size used by Contra-WiMAE, which we think can be attributed to overfitting. This gets particularly pronounced in the Deep-MIMO experiment. We also observe typical effects of overfitting during training, where the accuracy on the validation dataset degrades significantly after the first few epochs. *Does this resolve your concern?*
>
> Regarding the runtime comparison SpaRTran vs. Contra-WiMAE(original head): We report the average per-CSI-measurement latency in ms across 100 forward passes on the KUL dataset using an AMD Ryzen 9 9950X 16-Core Processor with a single thread. ContraWiMAE: Mean: 17.65 ms, Std: 0.2ms SpaRTran: Mean: 22.52 ms, Std:  0.14 ms. All methods are finetuned for 12h.
>
> However,  in our context, "simplicity" refers to the nature of the learned solutions. Specifically, our sparsity objective encourages sparse signal decompositions—solutions that explain the data with minimal active components.
> This notion aligns with principles such as Occam's razor, where "simple" characterizes the output structure rather than the model's computational or memory footprint. *Does this resolve your concern?*
>
> **Question 4. Which downstream model?**
> We employ ResNet-based heads for all downstream task evaluations across all baselines. The only exception is ContraWiMAE on the codebook selection task, where we cite linear probing results from Guler et al., as these outperform ResNet-based heads in the low-data regime (10%).
> We appreciate the reviewer's suggestion regarding linear probing, which is indeed a meaningful evaluation protocol for methods that compute representations over the full CSI. We appreciate the reviewer's suggestion regarding linear probing, which is indeed a meaningful protocol for methods computing representations over the full CSI—enabling direct modeling of inter-antenna correlations essential for environment-dependent tasks like wireless positioning.
>
> However, this evaluation does not directly transfer to our method due to a fundamental design difference: we compute per-link representations that are concatenated during finetuning. Theoretically, one cannot expect a combination of $N_r$ representations to exhibit a linear relationship to downstream targets, as independently linearly decomposing a signal does not generally imply joint linear decomposability.

---

> > ### Author Rebuttal · Reviewer_MDYm · 2026-04-01
> >
> > We thank the authors for their thoughtful responses and additional content, which address several of our concerns.
> >
> > Addressing each question:
> >
> > - **Q1 (Resolved):** Clear, thank you.
> >
> > - **Q2 (Resolved):** Understood. Regarding the $N_r$ sampling factor, we would recommend explicitly stating this detail in the manuscript along with your justification. Additionally, we believe that an ablation study where training steps are equalized (by reducing iterations by a factor of $N_r$) could provide valuable insights. This would address interesting questions such as: Is it more beneficial to train on diverse links or repeatedly on the same one? How do the learned representations differ when encoders are trained on different links? (We do not expect answers for these questions in the rebuttal).
> >
> > - **Q3 \& Q4 (Partially resolved)**: Thank you for the additional simulations.
> >     - **Follow-up question: You mention (Q4) that the downstream heads are ResNet-based, but are these identical (in terms of architecture, number of parameters) across all encoders/tasks (except the ContraWiMAE on the codebook selection task)?**
> >
> > We look forward to the authors' response.

---

> > > ### Author Response · Authors · 2026-04-01
> > >
> > > Thank you for your swift response! We are happy we could alleviate some of your questions and we think we can solve your additional question as well:
> > >
> > > About the follow up question:
> > > For all baselines we used the publicly available code and their respective ResNet head implementations and best performing head sizes described in the papers. This means the implementational details and number of parameters per head differ. We deliberately remained faithful to the original baseline implementations for two reasons:
> > >
> > > 1. Modifying baseline architectures could introduce confounding factors, making it difficult to attribute performance differences to the pretraining objectives. As stated above, we believe the reported architectures are already optimized for their respective methods.
> > > 2. We also believe, increasing downstream head capacity does not axiomatically improve results. The new experiments we provided in our rebuttal support this assumption.
> > >
> > > As we were limited by the character count in the rebuttal before we would like to use the space here to propose some improvements that may clear out some of the clarity issues you mentioned in the weaknesses part:
> > >
> > > - Add description of the quantity of datapoints seen during pretraining
> > > - Expand the discussion of single-link training compared to full CIS training
> > > - Add a table including the training details
> > > - Add downstream head sizes to Table 4.
> > >
> > > Thank you again for your time. We would be very appreciative if you could consider raising your score.

---

### Official Review · Reviewer_bnJm · 2026-03-11

**Soundness:** 2
**Presentation:** 2
**Significance:** 2
**Originality:** 2
**Overall Recommendation:** 3
**Confidence:** 4

**Summary:**

This paper incorporates domain knowledge into unsupervised representation learning for wireless channel state information (CSI) and proposes a new hybrid framework, SpaRTran, which leverages concepts from compressed sensing for wireless channel modeling.

**Compliance With Llm Reviewing Policy:**

Affirmed.

**Key Questions For Authors:**

The problem introduced in Section 3 is not clearly defined. After reading this section, it appears that the authors mainly provide background on CSI, but the specific problem that the paper aims to address is not clearly articulated.

In addition, the motivation for transitioning from Equation (2) to Equation (3) is unclear. In particular, the authors do not explain why it is necessary to convert Equation (2) into a sparse representation, or what advantages this transformation brings to the proposed method.

Furthermore, the proposed theoretical analysis does not provide sufficient insight into the novelty or performance of the proposed data-driven methods. For example, Remark 4.3 would benefit from first providing a clear technical explanation, followed by a more intuitive interpretation.

Finally, in the presentation of the proposed SpaRTran framework in Section 4, it is not clear how the design specifically emphasizes simplicity.

**Limitations:**

The authors should compare their results with existing systems that do not rely on AI-based approaches, in order to show whether the proposed method indeed provides improvements in simplicity and efficiency.

**Strengths And Weaknesses:**

Strengths

The perspective of integrating domain knowledge into the design of learning models to improve simplicity and efficiency is practical and well aligned with the needs of wireless communication systems.

The paper also provides strong theoretical analysis for the proposed model, which helps establish the foundation of the method.

In addition, the experimental setup and the downstream tasks used for evaluation are clearly illustrated and appear to be well designed.

Weaknesses

The problem statement is not clear.

The motivation by incorporating the wireless principles in not clear.

---

> ### Author Rebuttal · Authors · 2026-03-30
>
> First and foremost, thank you for your review. About the weaknesses: We think the mentioned weaknesses boil down to the key questions you ask and hope the unclarities are going to be resolved by our answers/revisions.
>
> > The problem introduced in Section 3 is not clearly defined. After reading this section, it appears that the authors mainly provide background on CSI, but the specific problem that the paper aims to address is not clearly articulated.
>
> The central goal of our work is to learn meaningful representations of the wireless channel without requiring labeled data during pretraining.
> We recognize that Section 3, as currently written, emphasizes the compressed sensing background in a way that may unintentionally obscure this primary objective. The role of compressed sensing in our framework is not to solve a traditional sparse recovery problem, but rather to provide a principled pretext task for self-supervised representation learning.
> We will revise Section 3 to state the representation learning objective more explicitly upfront and clarify how the compressed sensing formulation supports this goal. Thank you for raising this point!
>
> > In addition, the motivation for transitioning from Equation (2) to Equation (3) is unclear. In particular, the authors do not explain why it is necessary to convert Equation (2) into a sparse representation, or what advantages this transformation brings to the proposed method.
>
> We appreciate your comment regarding the presentation of compressed sensing concepts. The sparse channel representation in form of an underdetermined system of equations is central to our approach, as it embodies our key principle of simplicity—describing the signal with as few components as possible.
> While we agree that a more detailed introduction to compressed sensing fundamentals could benefit readers less familiar with the topic, we believe this falls outside the scope of a conference paper given the page constraints. To address this, we will add appropriate references to foundational compressed sensing literature, enabling interested readers to access the necessary background material. We hope this strikes a reasonable balance between accessibility and focus on our core contributions. *Does this resolve your concern?*
>
> > Furthermore, the proposed theoretical analysis does not provide sufficient insight into the novelty or performance of the proposed data-driven methods. For example, Remark 4.3 would benefit from first providing a clear technical explanation, followed by a more intuitive interpretation.
>
> The technical explanation is the proof in Appendix A, with the remark itself acting as the intuitive interpretation: The idea of the proof is simply that one can design an operator that takes a noisy signal which may live “dense” inside an infinite dimensional space, and projects it into a sparse space, under the assumption that such a sparse space exists.
> Theorems 4.1 and 4.2 act as a theoretical justification why a transformer that takes in an input signal $h(t)$ can produce a superior reconstruction compared to working on the native signal. Our core contribution remains the sparse-reconstruction objective for unsupervised pretraining.
>
> > Finally, in the presentation of the proposed SpaRTran framework in Section 4, it is not clear how the design specifically emphasizes simplicity.
>
> The simplicity bias in our approach is rooted in the principles of compressed sensing, which we adopt as the foundation for our pretext task. Following the principle of parsimony (Occam's Razor), we aim to describe the signal using as few components as possible while preserving all relevant information.
> This design philosophy is operationalized through three key components: (1) a gated sparse autoencoder, (2) a (learned) dictionary, and (3) a sparsity-inducing loss function. Together, these elements encourage the model to discover compact, physically meaningful representations.
> We will revise Section 4 to more explicitly highlight the connection between the principle of sparsity and our method's architectural design choices.
>
> Thank you for highlighting these unclarities. Could you confirm whether the above clarifications sufficiently address the issues you raised?

---

### Official Review · Reviewer_J897 · 2026-03-14

**Soundness:** 3
**Presentation:** 4
**Significance:** 3
**Originality:** 3
**Overall Recommendation:** 5
**Confidence:** 3

**Summary:**

The paper presents a hybrid model- and data-driven methodology for unsupervised representation learning of wires channel state information (CSI). The core idea is to incorporate the physics-informed sparse structure of the radio propagation environment directly into the architecture and training objective. The architecture combines a lightweight transformer-based encoder with a decoder inspired by gated sparse auto encoders. Another design choice is the pertaining on individual transmitter-receiver links instead of full CSI matrices, which makes the representation agnostic of the whole setup. These are then combined through a fine-tuning step. Experiements on three datasets exhibit consistent improvements over baselines. Accompanying theoretical analysis is also provided.

**Compliance With Llm Reviewing Policy:**

Affirmed.

**Final Justification:**

I thank the authors for their responses. I'm happy with these responses, and given the overall contribution of the paper, I increased my score.

**Key Questions For Authors:**

- Please clarify if the assumptions of your theoretical results are meaningful for the experiments you run. What is needed for them to hold? Please clarify whether the theoretical results really provide justification for the experimental results, or they are more of an intuition based on idealistic assumptions.

- Please include ablations with a simple autoencoder (without sparsity constraint) as an ablation study (with the same single link approach).

- How does this CS-inspired approach actually compare against the more traditional CS methods (e.g., OMP) assuming they have the same dictionary?

**Limitations:**

Yes

**Strengths And Weaknesses:**

Strengths:

Overall, this is an interesting and well-written paper which makes a novel (as far as I can tell) and valuable contribution to wireless signal learning. Incorporating the well-established sparsity of the radio frequency environment into the unsupervised pretraining objective, rather than directy transferring architectures and loss functions from NLP or vision to wireless problems (as is common in the literature) , is valuable. Experimental results are strong, especially in the more interesting and challenging domain-shift scenario and for the beamforming task. Theoretical analysis also complements well these numerical results by providing justification for why preconditioning with a learned operator improves sparse recovery rates. I also found the design choice of pertaining on individual transmitter-receiver links a smart and original choice. This way the learned representation is independent of the number of antennas or system topology. The benefits of this choice can be seen in the domain-shift results.

Even though I wasn't able to follow all the details of how the gating mechanism is combined with the phase generation, the general idea looks nice, and the implementation correct. (Wouldn't it be possible to directly generate complex values?)

Weaknesses:

- The performance is weaker in the low labeled-data ratio regime in Table 1. This is where unsupervised retraining should be most relevant. This is correctly attributed to the single-link pretrainign approach. This questions the validity of this approach.. This is probably the main limitation of the method, and should deserve more attention and discussion.

- More baselines and ablations can help better motivate the architectural choices.

---

> ### Author Rebuttal · Authors · 2026-03-30
>
> We first and foremost want to thank you for reviewing our paper.
> ## Regarding your outlined weaknesses:
>
> > The performance is weaker in the low labeled-data ratio regime in Table 1. This is where unsupervised pretraining should be most relevant. This is correctly attributed to the single-link pretrainign approach. This questions the validity of this approach. This is probably the main limitation of the method, and should deserve more attention and discussion.
>
> This point hits a subtle issue of our single-link pretraining approach. It is correct that our approach does not outperform the Contra-WiMAE baseline in the very low data regime (<10%) evaluated on the FH-IIS dataset. Nevertheless, SpaRTran achieves best performance in most of the cases in the very low data regime on the codebook selection task and shows superior domain shift capabilities while seeing only 5000 CSIs ($\approx 2$%) in case of the KUL dataset. However, we will revise our discussion to highlight this limitation. Does this resolve your issue?
>
> > More baselines and ablations can help better motivate the architectural choices.
>
> See answer to question 2.
>
> ## Regarding your questions:
> > 1. Please clarify if the assumptions of your theoretical results are meaningful for the experiments you run. What is needed for them to hold? Please clarify whether the theoretical results really provide justification for the experimental results, or they are more of an intuition based on idealistic assumptions.
>
> The core assumption is that we have a set of signals that can be represented in some reproducing kernel Hilbert space (which applies to all reasonable functions) and that the functions decompose into a signal $S$ and a noise component $\bar S$, which is a core assumption to make a reconstruction objective make sense in the first place.
> The main transfer from theory to practice is that the operator $O$ necessary for “rotating” the function $h$ into the improved coordinate frame may be of infinite size if we parameterize the transformation directly as a change-of-coefficients (since RKHS are, in general, infinite dimensional). We assume that the dense operator $O$ can be approximated by a transformer, which falls in line with the universal approximation theorem. In short, the theory proves existence of such an operator under very weak assumptions, while our transformer tries to realize it as a parametric model.
>
> > 2. Please include ablations with a simple autoencoder (without sparsity constraint) as an ablation study (with the same single link approach).
>
> We trained a simple autoencoder setup that uses a latent dimension of 32 with the single-link approach. We will replace Figure 3a. with the figure we shared here: https://anonymous.4open.science/r/temp-0B8C/ablation_cdf.pdf and discuss the new results accordingly. Does this answer your question?
>
> > 3. How does this CS-inspired approach compare against the more traditional CS methods (e.g., OMP) assuming they have the same dictionary?
>
> It is important to note that our approach and classical CS methods use sparse reconstruction for fundamentally different purposes. While our method employs sparse reconstruction as a pretext task for pretraining, CS methods solve the sparse reconstruction problem to directly obtain insights from the data (e.g., estimating the angle of arrival of an incoming signal) or compress it. After training the model the actual sparse decomposition is of limited interest since the transformer acts as feature extractor for our downstream tasks. As such, our method is much comparable to other pretraining tasks like masking, compared to sparse reconstruction schemes like OMP.
>
> Furthermore, a direct comparison of sparse decomposition quality is non-trivial: the number of captured multipath components may vary without necessarily affecting the quality of the learned representations. For these reasons, defining a fair and meaningful evaluation metric for this comparison remains an open challenge.
>
> Thanks again for your review. Please tell us if we missed any important points.

---

> > ### Author Rebuttal · Reviewer_J897 · 2026-04-03
> >
> > I thank the authors for their responses. I'm happy with these responses, and given the overall contribution of the paper, I increased my score.

---

### Decision · Program_Chairs · 2026-04-30

**Decision:**

Reject

**Comment:**

The problem statement is a somewhat well-known application of dictionary learning in wireless channel estimation. The work incorporates some new insights from sparse autoencoder to improve the performance. The reviewers are impressed by the results, but also point out that the theoretical analysis seems disconnected to the actual implementation. The main contribution is heavily inspired by Rajamanoharan et al. (2024), so its own originality is a bit limited.